# AdaptiVision: A Flexible and Efficient Vision Transformer for Adaptive Token Pruning

## Abstract

Transformer-based architectures have recently demonstrated remarkable performance in various vision tasks by capturing global contextual relationships through self-attention. However, this success comes at a high computational cost, as the self-attention mechanism scales quadratically with the number of visual tokens, limiting its scalability to high-resolution inputs and real-time applications. Although several recent efforts have aimed to reduce this complexity via token pruning or condensation, these methods often rely on heuristic importance scores or non-differentiable selection strategies, which can lead to suboptimal performance and lack of generalizability across tasks. To address these limitations, we propose AdaptiVision, a flexible and efficient Vision Transformer architecture designed to dynamically adapt the token set throughout the network. At the core of AdaptiVision is a differentiable token condensation module based on clustering, which groups semantically similar tokens and allows the model to retain only the most informative and representative ones while discarding redundancies. To guide this condensation process, we introduce a semantic guidance mechanism that incorporates auxiliary semantic signals (such as saliency or label-based cues) to preserve task-relevant structures during token reduction. Furthermore, we design auxiliary consistency and stability objectives that promote coherent token clustering across layers and inputs, enabling better generalization and robustness without sacrificing performance. We conduct extensive experiments across multiple challenging benchmarks to validate the effectiveness of our model. Notably, on the ImageNet-1K dataset, our proposed AdaptiVision achieves the highest Top-1 accuracy (79.8%) among comparable vision transformers while substantially reducing the number of parameters and FLOPs, demonstrating superior accuracy-efficiency trade-offs.

## 1 Introduction

Vision transformers (ViTs) have recently emerged as a powerful alternative to convolutional neural networks (CNNs) in a variety of computer vision tasks, including image classification Dosovitskiy et al. (2021a); Yuan et al. (2021), object detection Carion et al. (2020); Zhang et al. (2022), and semantic segmentation Strudel et al. (2021). Unlike CNNs, which exploit local spatial priors using shared convolutional filters, ViTs partition an image into a sequence of fixed-size patches and process them as tokens through global self-attention mechanisms. This architecture enables ViTs to model long-range dependencies effectively, resulting in strong performance across diverse benchmarks. However, the strength of ViTs comes at a significant computational cost: the self-attention operation has a quadratic complexity with respect to the number of tokens. Consequently, standard ViTs become prohibitively expensive for high-resolution images or real-time applications, where latency and efficiency are critical constraints.

A growing body of research has sought to reduce the token burden in ViTs through various compression, pruning, and routing mechanisms. One line of work focuses on token pruning strategies, such as DynamicViT Rao et al. (2021a) and Evo-ViT Xu et al. (2022), which dynamically drop less important tokens based on learned attention scores. While effective in reducing computation, these methods risk discarding semantically important tokens, particularly in early training stages where attention may be noisy or misaligned with task objectives. Other methods like TokenLearner Ryoo et al. (2021) and ToMe Bolya et al. (2023) propose learnable modules that merge or reweigh tokens,

reducing token count by generating a condensed representation. Although these techniques preserve more information than hard pruning, they often rely on heuristic or handcrafted criteria for merging and require careful tuning to balance performance and efficiency. Furthermore, approaches such as PVT Wang et al. (2021) and Swin Transformer Liu et al. (2021) adopt hierarchical structures to reduce token resolution at deeper stages, analogous to CNN pyramids, but at the cost of architectural complexity and limited modularity.

These baseline models highlight an essential challenge in ViT compression: the need for a principled mechanism that reduces token redundancy while preserving semantic content and being flexible enough to generalize across tasks. Most existing approaches rely solely on visual information to determine token importance, overlooking the fact that in many scenarios, external semantic cues, such as object annotations, spatial priors, or auxiliary modality embeddings, can offer valuable guidance in shaping token-level representations. However, incorporating such guidance into the token compression process remains largely unexplored. Moreover, few existing models enforce consistency or stability in token selection across augmentations or similar inputs, which can lead to erratic behavior and hinder training convergence.

Motivated by these limitations, we propose AdaptiVision, a flexible and efficient vision transformer framework that unifies dynamic token condensation, semantic guidance integration, and auxiliary supervision into a single, scalable architecture. The core component of AdaptiVision is a Dynamic Token Condensation (DTC) module that clusters input tokens into a set of representative super-tokens in a data-driven manner. Unlike pruning or merging strategies that rely on fixed importance scores, DTC performs differentiable soft clustering to preserve semantic fidelity while drastically reducing the token count. This allows AdaptiVision to adapt its granularity based on input complexity, retaining fine-grained detail where necessary and coarsening simpler regions.

To enhance token condensation, AdaptiVision incorporates a semantic guidance branch that injects external task-specific signals—such as class labels, masks, or learned embeddings—into the clustering process. These signals are projected into a latent space and integrated via cross-attention or fusion, enabling the model to leverage supervision when available while gracefully degrading when guidance is missing or noisy. Additionally, two auxiliary losses are employed: a clustering consistency loss to stabilize token-to-cluster assignments across augmentations, and a guidance alignment loss to align clusters with external semantic cues, improving generalization and robustness. Evaluations on ImageNet-1K, COCO, and ADE20K show that AdaptiVision achieves comparable or superior performance with fewer tokens and lower computational cost than baseline ViTs. In summary, our key contributions are:

1. We introduce AdaptiVision, a novel vision transformer framework that employs a differentiable token condensation strategy grounded in semantic clustering. This enables adaptive reduction of input tokens while maintaining essential semantic structures, significantly reducing computational overhead.

2. We introduce a semantic guidance mechanism that integrates auxiliary information into the condensation process, enhancing model flexibility.

3. We design principled auxiliary loss objectives that encourage stable and semantically meaningful token assignments across transformer layers. These objectives improve the model's generalization and consistency across various data distributions.

4. We conduct comprehensive experiments on multiple vision tasks to validate the effectiveness, efficiency, and generalizability of AdaptiVision compared to strong baselines.

## 2 RELATED WORKS

**Vision Transformers.** Transformer architectures, originally introduced for natural language processing tasks Vaswani et al. (2017), have demonstrated remarkable capability in modeling long-range dependencies through self-attention mechanisms, effectively replacing traditional recurrent networks. Motivated by this success, transformers have been adapted to computer vision, achieving competitive performance with convolutional neural networks (CNNs) in image classification Dosovitskiy et al. (2021a); Liu et al. (2021); Rao et al. (2021b); Zhou et al. (2021), object detection Carion et al. (2020), semantic segmentation Cheng et al. (2021); Zheng et al. (2021), and 3D vision tasks Yu et al. (2021).

The Vision Transformer (ViT) Dosovitskiy et al. (2021a) pioneered pure transformer-based image classification by treating image patches as tokens, although it required large-scale pretraining for competitive performance. Swin Transformer (Swin-T) Liu et al. (2021) introduced hierarchical feature maps and shifted windows to improve efficiency and scalability. More recent models, such as LeMeViT Zhang et al. (2024) and ToFe-LV Zhang et al. (2025), focus on lightweight and adaptive architectures to balance accuracy and computational cost. HiRED Arif et al. (2025) proposes hierarchical refinement for detailed feature extraction, while token-mixing strategies like ToMe Bolya et al. (2023) and DynamicViT Rao et al. (2021a) reduce computational overhead by selectively pruning tokens. These models collectively form strong baselines in modern vision transformer research.

**Token Reduction and Pruning.** Despite their effectiveness, vision transformers suffer from high computational costs due to the quadratic complexity of self-attention with respect to the number of tokens Dosovitskiy et al. (2021a). Recognizing that many tokens correspond to background or redundant information, several methods aim to reduce token count without sacrificing accuracy. DynamicViT Rao et al. (2021a) employs gating modules to discard less informative tokens, while TokenLearner Ryoo et al. (2021) condenses tokens using spatial attention, inspiring subsequent works such as TokenFuser Kim et al. (2024) and DART Yin et al. (2025). Adaptive token pruning methods, including AdaViT and A-ViT Yin et al. (2022), leverage attention entropy or learned binary masks to dynamically select important tokens. Evo-ViT Xu et al. (2022) gradually prunes tokens based on spatiotemporal similarity to maintain structural coherence. Reinforcement learning-based strategies, such as IA-RED$^2$ Pan et al. (2021), optimize pruning with interpretability-aware rewards. Finally, UViT Zeng et al. (2025) and EfficientFormer Li et al. (2022) integrate token reduction with architectural design to minimize FLOPs while preserving model performance.

## 3 METHOD

In this section, we present the architecture of AdaptiVision, designed for efficient and flexible visual representation through semantic-aware token processing and cross-modal integration. The model comprises three core components: Dynamic Token Clustering (DTC), AdaptiveFocus Attention (AFA), and Cross-Modal Token Fusion (CMTF), all operating on a ViT-compatible tokenized input.

Unlike conventional ViTs that tokenize images uniformly, AdaptiVision first partitions an image into $N$ non-overlapping patches, projects them into a $D$-dimensional embedding space, and adds positional encodings. The DTC module then clusters semantically similar tokens into super-tokens, reducing redundancy while preserving key information. These super-tokens are processed by AFA, which selectively attends to content-relevant regions, improving both generalization and computational efficiency. Appendix A provides the theoretical analysis.

### 3.1 INPUT PROCESSING

To prepare visual data for efficient and structured representation within the AdaptiVision, the input image $X \in \mathbb{R}^{H \times W \times C}$ is divided into a sequence of $N = \frac{H \times W}{P^2}$ non-overlapping patches of size $P \times P$. Each patch is flattened into a vector $x_i \in \mathbb{R}^{P^2 C}$ and projected into a $D$-dimensional embedding space using a learnable linear projection: $z_i = x_i W_p$, $W_p \in \mathbb{R}^{P^2 C \times D}$.

To preserve spatial structure, we incorporate positional encodings $E_{\text{pos}} \in \mathbb{R}^{N \times D}$, resulting in the initial token sequence:

$$Z_0 = [z_1, z_2, \ldots, z_N] + E_{\text{pos}}. \tag{1}$$

An optional $[CLS]$ token is prepended when performing classification tasks. Additionally, the architecture is designed to accommodate auxiliary tokens, such as those derived from external sensory signals or guidance inputs, which are appended after encoding. These auxiliary tokens can be selectively leveraged by later modules (e.g., CTF) for context-aware visual reasoning.

This input processing framework forms the foundation of AdaptiVision. A learnable linear projection maps raw image patches into compact, expressive embeddings, enhancing computational efficiency. Positional embeddings preserve spatial structure, essential for spatially sensitive tasks. The architecture also supports auxiliary tokens for incorporating contextual cues without altering

the main visual pipeline. This flexible design improves AdaptiVision's adaptability across a wide range of tasks.

## 3.2 Dynamic Token Clustering (DTC)

Dynamic token clustering (DTC) is a core component of the AdaptiVision architecture, designed to enhance computational efficiency by reducing the number of tokens processed in the transformer pipeline. Instead of relying on all $N$ input tokens, resulting from patch embedding, DTC compresses the token sequence into a significantly smaller set of $K \ll N$ super-tokens, thereby alleviating the quadratic computational cost of multi-head self-attention (MHSA).

To achieve this, we employ a differentiable soft $k$-means clustering mechanism. Each token $z_i \in \mathbb{R}^D$ is softly assigned to one of $K$ learned cluster centroids $\mu_k \in \mathbb{R}^D$. The assignment weight is defined as:

$$a_{ik} = \frac{\exp\left(-\|z_i - \mu_k\|^2/\tau\right)}{\sum_{j=1}^{K} \exp\left(-\|z_i - \mu_j\|^2/\tau\right)}, \tag{2}$$

where $\tau > 0$ is a temperature parameter that controls the sharpness of the assignments. The resulting super-tokens are computed as weighted aggregations:

$$s_k = \sum_{i=1}^{N} a_{ik} z_i, \tag{3}$$

forming the output token sequence $S = [s_1, \ldots, s_K] \in \mathbb{R}^{K \times D}$. Importantly, the value of $K$ can be dynamically adapted based on input complexity, such as token-level variance or entropy, to better reflect the semantic richness of different image regions.

DTC reduces self-attention complexity from $\mathcal{O}(N^2)$ to $\mathcal{O}(K^2)$, improving AdaptiVision's scalability and efficiency. Unlike static approaches such as Swin Transformer's fixed-window attention, DTC performs content-adaptive token aggregation, assigning more clusters to semantically rich regions and fewer to redundant areas. For each input, a complexity measure, such as the variance or entropy of token embeddings, is computed to capture heterogeneity. The cluster count $K$ is then dynamically scaled within a predefined range $[K_{\min}, K_{\max}]$ proportional to this complexity, ensuring richer inputs receive more clusters while simpler inputs use fewer. These cluster counts are used in a differentiable soft k-means procedure, producing variable-sized super-token sets that preserve important information in complex regions while maintaining computational efficiency.

## 3.3 AdaptiveFocus Attention (AFA)

AdaptiveFocus Attention (AFA) augments the standard MHSA mechanism by introducing a context-aware modulation strategy, enabling more dynamic and input-sensitive attention. By leveraging global contextual information, AFA incorporates a modulation matrix that adjusts attention weights based on the overall characteristics of the input, allowing the model to effectively prioritize salient and task-relevant regions.

Given a token sequence $S_l \in \mathbb{R}^{K \times D}$ at layer $l$, standard MHSA computes the query, key, and value matrices:

$$\mathcal{Q} = S_l W_Q, \quad \mathcal{K} = S_l W_K, \quad \mathcal{V} = S_l W_V. \tag{4}$$

AFA enhances this by introducing a modulation matrix $M \in \mathbb{R}^{K \times K}$, predicted by a lightweight feedforward network $f_m$ conditioned on global contextual information:

$$G = \text{Pool}(S_l), \quad M = f_m(G), \tag{5}$$

where $G \in \mathbb{R}^D$ is a pooled global descriptor (e.g., mean-pooled token embeddings).

The attention scores are modulated as follows:

$$\tilde{A}_j = \text{softmax}\left(\frac{\mathcal{Q}_j \mathcal{K}_j^T}{\sqrt{d_k}} \cdot M\right) \mathcal{V}_j, \tag{6}$$

where $M \in \mathbb{R}^{K \times K}$ is the context-dependent modulation matrix that adjusts the attention distribution.

The final output of the attention layer is computed by concatenating the heads, applying a linear projection, and adding a residual connection followed by layer normalization:

$$S_{l+1} = \text{LayerNorm}\left(S_l + \text{Concat}(\tilde{A}_1, \ldots, \tilde{A}_h)W_O\right). \tag{7}$$

AFA dynamically modulates attention weights using global statistics $G$, allowing the model to emphasize task-relevant regions and suppress less informative content, unlike standard ViTs, which treat all tokens equally. This results in more selective and robust attention. The lightweight, fully differentiable design enables seamless integration and end-to-end training, enhancing AdaptiVision's generalization, feature discrimination, and cross-domain robustness.

### 3.4 CROSS-MODAL TOKEN FUSION (CMTF)

The cross-modal token fusion (CMTF) module facilitates the integration of auxiliary guidance information, such as text embeddings, depth cues, or other contextual signals, into the token stream of the visual backbone. Given auxiliary tokens $G_t \in \mathbb{R}^{M \times D}$ and the super-token sequence $S_l \in \mathbb{R}^{K \times D}$ at layer $l$, a cross-attention mechanism is applied to align and fuse these representations:

$$\mathcal{Q} = S_l W_{Q_c}, \quad \mathcal{K} = G_t W_{K_c}, \quad \mathcal{V} = G_t W_{V_c}, \tag{8}$$

$$A_c = \text{softmax}\left(\frac{\mathcal{Q}\mathcal{K}^T}{\sqrt{d_k}}\right)\mathcal{V}, \tag{9}$$

where $W_{Q_c}, W_{K_c}, W_{V_c} \in \mathbb{R}^{D \times d_k}$ are learnable projection matrices.

To regulate the contribution of auxiliary information, a gating mechanism is introduced:

$$S_l' = S_l + \sigma(A_c W_g), \tag{10}$$

where $W_g \in \mathbb{R}^{d_k \times D}$ is a trainable weight matrix and $\sigma$ denotes the sigmoid activation function.

This mechanism enables the network to dynamically integrate auxiliary guidance into the visual token representation while preserving the simplicity of the core architecture. In contrast to conventional approaches that rely on complex multi-branch designs, CMTF provides a lightweight and efficient solution that enhances the model's adaptability across a wide range of tasks. Moreover, the inclusion of a gating mechanism ensures that the influence of external signals is effectively regulated, allowing the model to maintain robust performance even in the absence of auxiliary inputs.

### 3.5 OUTPUT STAGE

The final output stage of the model is designed for flexibility and efficiency across a wide spectrum of vision tasks. For classification, a compact global representation is extracted from either the $[CLS]$ token or the pooled set of super-tokens $S_L$ at the final layer:

$$z_{\text{out}} = \text{Pool}(S_L)$$
$$y = W_o \cdot \text{LayerNorm}(z_{\text{out}}) \tag{11}$$

This stage seamlessly supports both standard classification and more complex downstream tasks, such as object detection or semantic segmentation, by appending task-specific heads. By operating on the reduced token set produced by the DTC module, the output stage ensures computational efficiency without compromising representational quality. The use of layer normalization and a final linear projection preserves compatibility with established ViTs, allowing AdaptiVision to remain robust and adaptable across a wide variety of vision applications.

### 3.6 TRAINING OBJECTIVE

The training objective of the proposed AdaptiVision framework is formulated to jointly optimize three aspects critical to robust performance across vision tasks: task-specific accuracy, structural

consistency of token representations, and alignment with auxiliary modalities when available. The total loss function is defined as:

$$\mathcal{L} = \mathcal{L}_{\text{CE}} + \lambda_1 \mathcal{L}_{\text{cluster}} + \lambda_2 \mathcal{L}_{\text{align}}, \tag{12}$$

where $\mathcal{L}_{\text{CE}}$ denotes the standard cross-entropy loss used for classification tasks, ensuring that the model learns accurate label predictions based on the final representation. Given the predicted class probabilities $\hat{y}_i$ and ground truth labels $y_i$, this loss is computed as:

$$\mathcal{L}_{\text{CE}} = -\sum_{i=1}^{N} y_i \log(\hat{y}_i). \tag{13}$$

The second component, $\mathcal{L}_{\text{cluster}}$, is a clustering consistency loss that enhances the stability and semantic coherence of the token grouping process within the DTC module. It ensures that tokens $z_i$ are tightly clustered around their respective cluster centroids $\mu_k$, encouraging structurally meaningful token representations. It is defined as:

$$\mathcal{L}_{\text{cluster}} = \sum_{k=1}^{K} \sum_{i=1}^{N} a_{ik} \left\| z_i - \mu_k \right\|^2, \tag{14}$$

where $a_{ik}$ is an assignment coefficient indicating the degree to which token $z_i$ belongs to cluster $k$, $\mu_k$ is the centroid of cluster $k$, and $N$ is the total number of tokens.

The third component, $L_{\text{align}}$, is an optional alignment loss that encourages visual representations to align with auxiliary inputs (e.g., text or depth maps). Based on contrastive learning, it pulls positive pairs—consisting of the pooled visual output $z_{\text{out}}$ and its corresponding auxiliary embedding $g_t$—closer in the embedding space, while pushing negative pairs $g_t'$ from other instances apart. Formally, the loss maximizes similarity for $(z_{\text{out}}, g_t)$ and minimizes it for $(z_{\text{out}}, g_t')$, using a similarity metric such as cosine similarity with temperature scaling to control the sharpness of the distribution. Mathematically:

$$L_{\text{align}} = -\log \frac{\exp(\text{sim}(z_{\text{out}}, g_t)/\tau)}{\exp(\text{sim}(z_{\text{out}}, g_t)/\tau) + \sum_{g_t'} \exp(\text{sim}(z_{\text{out}}, g_t')/\tau)}, \tag{15}$$

where $\tau$ is a temperature hyperparameter and $\text{sim}(\cdot, \cdot)$ denotes the similarity function. This formulation ensures the model learns semantic alignment across modalities by explicitly contrasting relevant positive pairs against semantically unrelated negatives sampled in-batch.

The hyperparameters $\lambda_1$ and $\lambda_2$ control the contributions of clustering consistency and alignment with auxiliary guidance, respectively. Together, these terms encourage AdaptiVision to learn stable and discriminative representations while flexibly incorporating external signals when available, improving adaptability and performance across diverse vision tasks.

## 4 EXPERIMENT

In this section, we extensively evaluate the proposed AdaptiVision framework across multiple standard vision benchmarks to assess its effectiveness, efficiency, and generalizability.

### 4.1 DATASETS

To evaluate the effectiveness and generalizability of AdaptiVision, we conduct experiments across a diverse set of benchmark datasets spanning multiple vision tasks. For image classification, we use CIFAR-10, CIFAR-100, Tiny ImageNet, and the large-scale ImageNet-1K. We also include the Chest X-ray dataset to assess performance in a medical imaging context. For object detection, we use the COCO2017 dataset, which contains complex scenes with multiple objects. For semantic segmentation, we utilize ADE20K, a challenging dataset with 150 classes and dense annotations.

### 4.2 IMPLEMENTATION DETAILS

Our proposed AdaptiVision framework is implemented using the PyTorch library. All models are trained on NVIDIA A4500 GPUs with mixed-precision training enabled for efficiency. For consistent evaluation, we compare AdaptiVision with a range of competitive baseline models, including

ViT Dosovitskiy et al. (2021b), Swin-T Liu et al. (2021), LeMeViT Zhang et al. (2024), ToFe-LV Zhang et al. (2025), HiRED Arif et al. (2025), ToMe Bolya et al. (2023) and DynamicViT Rao et al. (2021a), following the same training and evaluation protocols as used in their original implementations. We use the AdamW optimizer with a cosine annealing learning rate schedule. The base learning rate is set to 5e-4, and weight decay to 0.05. All models are trained for 300 epochs.

## 4.3 COMPARISON WITH STATE-OF-THE-ART METHODS ON MULTIPLE BENCHMARKS

Table 1 showcases a detailed comparison of the proposed AdaptiVision model across four diverse datasets: CIFAR-10, CIFAR-100, TinyImageNet, and ChestX-ray. The evaluation is performed using Top-1 accuracy, Top-5 accuracy, and throughput (measured in images per second). Across all datasets, AdaptiVision consistently achieves superior performance while maintaining lower computational complexity in terms of parameters and FLOPs. On CIFAR-10, AdaptiVision attains a Top-1 accuracy of 98.9% and a Top-5 accuracy of 99.9%, which are the highest among all compared methods. Additionally, it records the highest throughput at 712 images per second, highlighting its efficiency. This performance trend continues on the CIFAR-100 dataset, where AdaptiVision achieves a Top-1 accuracy of 89.7% and a Top-5 accuracy of 99.5%, again outperforming all baselines. It also maintains the highest throughput at 700 images per second, demonstrating its ability to scale across datasets of increasing complexity. Similarly, on the TinyImageNet dataset, AdaptiVision delivers a Top-1 accuracy of 79.2% and a Top-5 accuracy of 95.0%, outperforming existing models while maintaining a strong inference speed of 685 images per second. On the ChestX-ray dataset, which involves real-world medical imaging, the model achieves the highest classification performance with a Top-1 accuracy of 95.1%, further emphasizing its robustness and applicability to critical domains. Overall, AdaptiVision demonstrates a favorable balance between accuracy and computational efficiency, setting a new benchmark in adaptive vision transformer design.

Table 1: Comparison of AdaptiVision with state-of-the-art models across multiple datasets in terms of Top-1 Accuracy, Top-5 Accuracy, and Throughput (images/second) (Thpt).

| Model | Params (M) | FLOPs (G) | CIFAR-10 | | | CIFAR-100 | | | TinyImageNet | | | ChestX-ray (Top-1) |
|---|---|---|---|---|---|---|---|---|---|---|---|---|
| | | | Top-1 | Top-5 | Thpt | Top-1 | Top-5 | Thpt | Top-1 | Top-5 | Thpt | (%) |
| ViT | 86.0 | 17.6 | 98.0 | 99.8 | 475 | 87.1 | 98.9 | 460 | 74.2 | 93.0 | 450 | 92.6 |
| Swin-T | 28.3 | 4.5 | 98.3 | 99.8 | 624 | 88.0 | 99.1 | 610 | 76.2 | 93.7 | 598 | 93.4 |
| DynamicViT | 27.0 | 3.4 | 98.4 | 99.8 | 655 | 88.3 | 99.2 | 643 | 77.1 | 94.0 | 625 | 94.0 |
| LeMeViT | 30.0 | 4.2 | 98.5 | 99.8 | 640 | 88.5 | 99.2 | 622 | 77.4 | 94.2 | 608 | 94.2 |
| ToFe-LV | 26.0 | 3.3 | 98.6 | 99.9 | 670 | 89.0 | 99.3 | 655 | 78.1 | 94.4 | 640 | 94.4 |
| HiRED | 24.5 | 3.4 | 98.7 | 99.9 | 685 | 89.2 | 99.4 | 670 | 78.3 | 94.6 | 652 | 94.5 |
| ToMe | 25.8 | 3.1 | 97.8 | 99.5 | 618 | 88.4 | 98.6 | 620 | 77.8 | 94.8 | 663 | 93.6 |
| **AdaptiVision (Ours)** | **18.2** | **3.1** | **98.9** | **99.9** | **712** | **89.7** | **99.5** | **700** | **79.2** | **95.0** | **685** | **95.1** |

Table 2: Top-1 Accuracy (%) and Throughput on ImageNet-1K

| Model | Params (M) | FLOPs (G) | Top-1 Acc (%) |
|---|---|---|---|
| ViT | 86 | 17.6 | 77.9 |
| Swin-T | 28.3 | 4.5 | 78.4 |
| DynamiViT | 27.0 | 3.4 | 78.6 |
| LeMeViT | 30 | 4.2 | 77.9 |
| ToFe-LV | 26 | 3.3 | 79.2 |
| HiRED | 24.5 | 3.4 | 78.9 |
| ToMe | 25.8 | 3.9 | 78.8 |
| AdaptiVision (ours) | 18.2 | 3.1 | 79.8 |

## 4.4 IMAGE CLASSIFICATION ON IMAGENET-1K

Table 2 presents a detailed comparison of the proposed AdaptiVision model against several state-of-the-art Transformer-based vision architectures on the ImageNet-1K dataset. With only 18.2M parameters and 3.1 GFLOPs, AdaptiVision achieves the highest Top-1 Accuracy of 79.8%, outperforming recent efficient models such as ToFe-LV (79.2%, 3.3G FLOPs), HiRED (78.9%, 3.4G FLOPs), and DynamicViT (78.6%, 3.4G FLOPs), while using fewer parameters and lower computational cost. Compared to Swin-T (78.4%, 4.5G FLOPs) and LeMeViT (77.9%, 4.2G FLOPs), AdaptiVision not only improves accuracy by a clear margin but also operates with significantly lower FLOPs. Even against the original ViT model, which has 86M parameters and 17.6 GFLOPs,

AdaptiVision provides a more efficient and lightweight alternative with better accuracy. These results highlight the effectiveness of AdaptiVision in balancing performance and computational efficiency, making it a strong candidate for high-performance image classification tasks. The Fig. 1 (a) illustrates the performance of our proposed AdaptiVision model on the ImageNet-1K dataset under varying FLOPs. AdaptiVision consistently outperforms existing methods like DynamicViT and ToFe-LV, especially in low-computation settings, demonstrating superior accuracy-efficiency trade-offs.

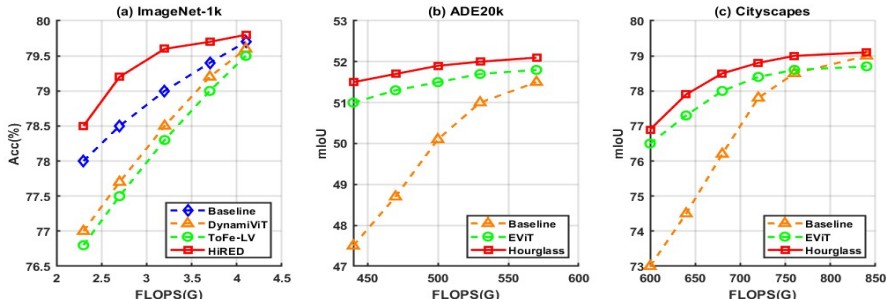

Figure 1: Comparison with previous models across three tasks under varying FLOPs constraints. (a) ImageNet-1K for classification, (b) ADE20K for segmentation, and (c) Cityscapes for segmentation. All methods follow identical token reduction settings. Our model consistently achieves superior performance, especially under aggressive token compression.

## 4.5 Semantic Segmentation

We extend our evaluation to semantic segmentation tasks using the Segmenter-L/16 architecture as the baseline. Our method, AdaptiVision, is integrated into the Vision Transformer block of Segmenter. For a fair comparison, we evaluate our model against existing token selection methods, including Hourglass Liang et al. (2022a) and EViT Liang et al. (2022b), under varying FLOPs configurations.

As illustrated in Fig. 1 (b) (ADE20k) and Fig. 1 (c) (Cityscapes), AdaptiVision consistently achieves superior performance across a wide range of computational budgets. Notably, it maintains higher mIoU with lower FLOPs, demonstrating the efficiency and effectiveness of our token selection strategy. For instance, on ADE20k, AdaptiVision preserves up to 30% FLOPs reduction with negligible performance loss, outperforming both EViT and Hourglass. On Cityscapes, it exceeds Hourglass by 0.4% mIoU (78.8 vs. 78.4) under equivalent FLOPs, while offering a 25% lossless compression. These results highlight the strength of AdaptiVision in dense prediction tasks. Unlike token-exclusive methods such as EViT, our many-to-many adaptive token preservation mechanism retains critical spatial-semantic information, especially in lower-compression regimes. This capability plays a pivotal role in sustaining model accuracy under constrained computational settings, proving the robustness and adaptability of AdaptiVision for semantic segmentation.

Table 3: Detailed comparison for object detection on COCO.

| Model | Backbone | GFLOPs ↓ | FPS ↑ | mAP ↑ |
|---|---|---|---|---|
| DINO | Swin-L | 936 | 4.74 | 58.5 |
| +Hourglass | Swin-L | 746 | 5.64 | 56.9 |
| +AdaptiVision (Ours) | Swin-L | 734 | 5.72 | 57.3 (+0.4) |

## 4.6 Object Detection on COCO

We evaluate AdaptiVision on object detection by integrating it with the Swin-L Zhang et al. (2022) backbone in the DINO Zhang et al. (2022) detector. Unlike classification tasks, object detection does not use class tokens, making many existing dynamic token pruning methods inapplicable. Therefore, we compare AdaptiVision with the state-of-the-art Hourglass Liang et al. (2022a) under identical compression settings.

As reported in Table 3, AdaptiVision achieves a mAP of 57.3%, outperforming Hourglass by +0.4% while also reducing GFLOPs and improving inference speed (FPS). These results demonstrate that our matrix-efficient token selection effectively preserves performance in dense prediction tasks and validate AdaptiVision's scalability and generalizability to transformer-based backbones like Swin-L.

## 4.7 ABLATION STUDY

We conduct a thorough ablation study on the image classification task to assess the effectiveness of our proposed AdaptiVision framework and analyze the impact of key hyperparameters. We also present the generalization to out-of-distribution (OOD) experiments and the evaluation of individual model components in Appendix B.

**Effect of Temperature ($\tau$) in Token Transformation:** We start by analyzing the impact of the temperature parameter $\tau$ in the token transformation module on ImageNet-1K classification performance. As presented in Table 4, the classification accuracy steadily improves as the temperature increases from 1 to 150, reaching peak performance at $\tau = 150$. Beyond this point, the improvement plateaus, and performance slightly declines, indicating a saturation point in model sensitivity.

This trend supports our hypothesis: a very low temperature introduces excessive noise and weakens discriminative capacity by preserving less informative tokens, while excessively high temperatures tend to oversmooth the transformation, potentially filtering out relevant features. The optimal range of $\tau$ lies between 100 and 200, demonstrating the robustness and stability of AdaptiVision to a broad range of parameter settings.

Table 4: Influence of temperature $\tau$ on classification accuracy (%) for AdaptiVision.

| Temperature $\tau$ | 1 | 20 | 100 | 150 | 200 | 250 |
|---|---|---|---|---|---|---|
| Accuracy (%) | 74.5 | 79.2 | 79.5 | 79.7 | 79.5 | 79.4 |

Table 5: Ablation study on the effect of token reduction strategy in AdaptiVision on ImageNet-1K.

| Model | Acc (%) | FLOPs (G) |
|---|---|---|
| No pruning | 78.0 | 4.6 |
| Random pruning | 77.8 | 3.1 |
| Importance-based (DynamicViT) | 78.4 | 3.2 |
| Semantic-aware (Ours) | 79.8 | 3.1 |

**Effect of Token Reduction Strategy in AdaptiVision:** To evaluate the effectiveness of the token reduction strategy in our proposed AdaptiVision model, we conduct a comprehensive ablation study using ImageNet-1K. Table 5 presents a comparative analysis of various token reduction strategies, including no pruning, random pruning, importance-based pruning (as used in DynamicViT), and our proposed semantic-aware token condensation method.

Our semantic-aware token condensation method achieves the highest accuracy while maintaining lower FLOPs, outperforming both random and importance-based pruning strategies. This demonstrates that preserving semantically meaningful tokens leads to more informative and efficient representations, highlighting the superiority of the AdaptiVision token reduction approach.

## 5 CONCLUSION

In this paper, we presented AdaptiVision, a flexible and efficient vision transformer framework designed to tackle the computational overhead and semantic redundancy inherent in standard Vision Transformer (ViT) architectures. Unlike prior token reduction approaches that depend on heuristic scoring or class-token attention, AdaptiVision introduces a principled, semantic-aware token condensation mechanism that enables adaptive pruning based on both content relevance and structural consistency. To further strengthen its performance, we incorporate an auxiliary guidance branch and design auxiliary objectives that enforce clustering stability and semantic alignment, ensuring robust token selection and better generalization across diverse tasks. Our method achieves state-of-the-art results while reducing computational cost, demonstrating strong scalability and adaptability. Overall, AdaptiVision provides a task-agnostic and semantically guided blueprint for building efficient transformer-based vision models, paving the way for future advances in lightweight and high-performance visual understanding.

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

## A  THEORETICAL ANALYSIS

To formally establish the computational advantage of our proposed DTC module, we present the following theorem, which quantifies the reduction in computational complexity achieved by AdaptiVision's attention mechanism.

**Theorem 1.** *Let $N$ be the number of input tokens and $K$ denote the number of super-tokens produced by the DTC module, where $K \ll N$. The computational complexity of the MHSA mechanism in ViTs is $O(N^2 D)$, where $D$ is the token embedding dimension. AdaptiVision reduces this complexity to $O(K^2 D)$, thereby offering a substantial gain in efficiency.*

*Proof.* In a conventional ViT, each MHSA block operates on the full sequence of $N$ tokens. Computing the attention score matrix $QK^T \in \mathbb{R}^{N \times N}$ across $h$ heads incurs a computational cost of $O(N^2 D)$, which becomes a bottleneck as $N$ grows.

AdaptiVision addresses this by applying DTC, which clusters the input tokens into $K$ semantically coherent super-tokens. The MHSA now operates on this reduced set, and the attention score computation $QK^T \in \mathbb{R}^{K \times K}$ requires only $K^2 D$ operations. Even though the clustering step requires $O(NKD)$ operations for computing assignment weights and cluster centroids, this cost is significantly smaller than $O(N^2 D)$ when $K \ll N$, making it negligible in the overall computational cost. □

This theorem underpins the scalability of AdaptiVision by showing that DTC reduces attention complexity to $O(K^2 D)$, enabling efficient processing of longer sequences or high-resolution images without loss of expressiveness. This efficiency allows AdaptiVision to perform competitively even in resource-constrained or real-time settings.

## B  ADDITIONAL ABLATION STUDY

**Generalization to Out-of-Distribution (OOD):** To assess the robustness of our proposed AdaptiVision model against distributional shifts, we evaluate its performance on two challenging OOD benchmarks: ImageNet-A and ImageNet-R. As shown in Table 6, AdaptiVision achieves notable improvements over baseline Vision Transformer models, demonstrating its enhanced ability to generalize beyond the training distribution. This superior performance highlights the benefit of our semantic-aware learning framework in improving model robustness under real-world conditions.

Table 6: Generalization to Out-of-Distribution (OOD) datasets: ImageNet-A and ImageNet-R

| Model | ImageNet-A (%) | ImageNet-R (%) |
|---|---|---|
| ViT | 41.0 | 62.8 |
| DeiT | 43.7 | 64.5 |
| AdaptiVision (ours) | 47.1 | 67.2 |

Table 7: Ablation study results on ImageNet-1K. Including each component improves accuracy and/or efficiency.

| Model Variant | DTC | AFA | CMTF | Top-1 Acc (%) | FLOPs (G) |
|---|---|---|---|---|---|
| Baseline ViT | ✗ | ✗ | ✗ | 77.9 | 17.6 |
| + DTC | ✓ | ✗ | ✗ | 78.6 | 3.8 |
| + DTC + AFA | ✓ | ✓ | ✗ | 79.2 | 4.0 |
| + DTC + AFA + CMTF (Full) | ✓ | ✓ | ✓ | 79.8 | 3.1 |

**Evaluation of Individual Model Components:** To quantify the individual contributions of the proposed modules, Dynamic Token Clustering (DTC), AdaptiveFocus Attention (AFA), and Cross-Modal Token Fusion (CMTF), we conduct a comprehensive ablation study on the ImageNet-1K dataset. Starting from a baseline vision transformer lacking these components, we progressively add each module and evaluate their impact on Top-1 accuracy, computational complexity (FLOPs),

and inference throughput. This analysis highlights the complementary benefits and computational trade-offs of each component.

As shown in Table 7, the baseline ViT achieves a Top-1 accuracy of 77.9% but at a high computational cost of 17.6 GFLOPs. Incorporating DTC drastically reduces FLOPs to 3.8 G, a nearly 80% reduction, while also increasing accuracy to 78.6% by effectively condensing redundant tokens. Adding AFA further improves accuracy to 79.2% by adaptively modulating attention, with only a minor increase in FLOPs to 4.0 G. Finally, the full model with CMTF reaches the highest Top-1 accuracy of 79.8%, demonstrating the benefit of integrating cross-modal semantic information, while maintaining an efficient FLOPs of 3.1 G due to the synergy between DTC and CMTF.

