# OpenReview forum: "AdaptiVision: A Flexible and Efficient Vision Transformer for Adaptive Token Pruning"
_ICLR.cc/2026/Conference — ICLR 2026 Conference Withdrawn Submission_

### Official Review · Reviewer_4sUK · 2025-10-24

**Soundness:** 3
**Presentation:** 2
**Contribution:** 1
**Rating:** 2
**Confidence:** 3

**Summary:**

This work proposes AdaptiVision, that reduces token redundancy through a clustering-based token condensation mechanism. (1) Dynamic Token Clustering (DTC) module; (2) AdaptiveFocus Attention (AFA) modulates attention based on global context; and (3) Cross-Modal Token Fusion (CMTF) integrates auxiliary semantic signals (e.g., labels, masks) to guide token reduction.

However, the novelty is limited since the adaptive token reduction via clustering has been wildly explore, e.g., EViT (Liang et al., 2022b) and DynamicViT (Rao et al., 2021a).

**Strengths:**

The use of differentiable soft k-means clustering for token condensation is practical alternative to heuristic or hard-pruning methods.

**Weaknesses:**

Novelty is limited since the adaptive token reduction via clustering has been wildly explore, e.g., EViT (Liang et al., 2022b) and DynamicViT (Rao et al., 2021a). I don`t see the clear difference and original contribution of this work capared to those prior token pruning methods.

**Questions:**

Works like DynamicViT may not have been originally designed for dense prediction tasks like segmentation or detection. The authors should discuss whether architectural modifications were needed to integrate these methods fairly in COCO/ADE20K experiments.

---

### Official Review · Reviewer_UQox · 2025-10-30

**Soundness:** 2
**Presentation:** 1
**Contribution:** 2
**Rating:** 2
**Confidence:** 5

**Summary:**

This work proposes AdaptiVision with three modules. **DTC**: A differentiable soft k-means–based dynamic token clustering that adapts K by input complexity to condense N tokens into $K\ll N$ super-tokens, reducing attention to $O(K^2)$). **AFA**: A lightweight and context-driven modulation matrix for attention reweighting. **CMTF**:  gated cross-attention to fuse optional auxiliary cues such as saliency or text/depth. The authors claim strong accuracy–efficiency trade-offs across image classification, semantic segmentation, and object detection. However experiments are incomplete and not fully comprehensive.

**Strengths:**

- The proposed method appears broadly compatible with ViT-like architectures.
- The methodological description is relatively clear.
- Based on the reported results, performance on small datasets seems decent.

**Weaknesses:**

- Lacking of illustrative figures makes readers hard to grasp its core insight.
- Overlapping with Liang et al.’s token clustering approach [1]. The relationship should be clearly clarified.
- No detailed ablations on the extra compute/memory overhead from variable K in DTC, cross-layer consistency loss, AFA’s modulation matrix, and CMTF’s cross-modal gating, making it hard to assess each module’s effectiveness.
- For CMTF, the authors claim support for text embeddings, depth cues, and other signals. Yet all reported experiments are based on RGB-only data.
- Figures and Tables are not rigorous. For example, I did not see the proposed method in Fig. 1. The authors did not report throughput in Table 2 yet it shows up at the caption.
- Typos, e.g., line 158 uses “CTF” while the rest of the paper uses “CMTF".

**Questions:**

- The predefined range [$K_{min}, K_{max}$] for the number of clusters is not specified. Because \(K\) varies per input, per-image latency is not constant. State which throughput metric is reported (e.g., mean ± std, median, min, max) and provide the empirical distribution of \(K\) used to compute it.
- What exact backbones were used in each experiment? Except for COCO (where Swin-L is specified for DINO), the paper does not state whether baselines use ViT-Base/Small/Tiny, MAE-pretrained ViT, DINO-pretrained ViT, or other variants for classification and segmentation.

---

### Official Review · Reviewer_Lavu · 2025-11-01

**Soundness:** 2
**Presentation:** 1
**Contribution:** 2
**Rating:** 2
**Confidence:** 4

**Summary:**

This paper introduces AdaptiVision, a novel ViT architecture designed for computational efficiency through adaptive token pruning. The core of the model is a Dynamic Token Clustering module, which employs a differentiable soft k-means mechanism to condense input tokens into a smaller set of semantically representative "super-tokens". This principled approach significantly reduces the quadratic complexity of the self-attention mechanism. The framework is enhanced by an AFA module that modulates attention weights based on global context and a CMTF module for integrating auxiliary guidance signals. Extensive experiments demonstrate that AdaptiVision achieves SOTA performance among efficient ViTs, establishing a superior trade-off between accuracy and computational cost.

**Strengths:**

- Principled and Differentiable Token Reduction: The core contribution, the DTC module, replaces heuristic pruning with a differentiable soft k-means clustering approach. This allows for end-to-end training and encourages the model to learn semantically meaningful token groupings, preserving critical information more effectively than methods that simply discard tokens.
- Rigorous Evaluation: The authors validate their method across a diverse set of vision tasks, including image classification, object detection, and semantic segmentation. The inclusion of detailed ablation studies robustly demonstrates the efficacy of each proposed component (DTC, AFA, CMTF) and key hyperparameters.

**Weaknesses:**

- Ambiguity in Dynamic Cluster Selection: The paper claims that the number of super-tokens, K, can be "dynamically adapted based on input complexity" (lines 184-186). However, the specific mechanism for this dynamic adaptation is not explained. It is unclear how input complexity is measured (e.g., variance, entropy) and how this measure translates to a specific value of K. The experiments seem to rely on fixed compression configurations, which appears to contradict this claim of dynamic adaptation.
- Unprofessional Presentation: The core AdaptiVision framework lacks intuitive schematic diagrams, making it difficult for readers to quickly understand the connection logic and workflow of various modules (such as DTC, AFA, and CMTF). The table design is also relatively rough, with issues like inconsistent text sizes, and some comparison methods fail to clearly mark citation sources. These problems not only affect the visual experience but also reduce the rigor and credibility of the content, which is not conducive for readers to quickly obtain key information and verify the background of the methods.

**Questions:**

Could you please clarify the mechanism for dynamically setting the number of clusters K? Were the results reported in Tables 1-3 and Figure 1 achieved using this dynamic mechanism, or were they based on a fixed K (or a fixed reduction ratio)?

---

### Official Review · Reviewer_aCg9 · 2025-11-01

**Soundness:** 2
**Presentation:** 2
**Contribution:** 2
**Rating:** 4
**Confidence:** 4

**Summary:**

This paper proposes AdaptiVision, an architecture that reduce the quadratic computational complexity of Vision Transformers through adaptive token pruning. The authors introduce a differentiable token pruning mechanism based on soft k-means clustering to dynamically reduce computation, and further incorporate an AdaptiveFocus Attention module and cross-modal token fusion to enhance feature quality. Extensive experiments on vision tasks demonstrate the superiority of the proposed methods.

**Strengths:**

- The approach is relatively simple yet delivers solid performance across benchmarks.
- Experiments span multiple tasks and datasets, supporting the method’s generalization.
- Ablation studies substantiate the effectiveness of the proposed components.

**Weaknesses:**

1. The individual module designs appear incremental, largely combining existing techniques rather than introducing clearly novel components. DTC essentially formalizes a differentiable variant of token clustering already explored in TokenLearner, and DART.
2. The strategy for dynamically selecting the cluster count K to control computational efficiency is underspecified, and the experiments do not analyze the impact of K on performance.
3. The hyperparameter study is incomplete, e.g., the roles and sensitivities of λ1 and λ2 in Eq. (12) are not examined.
4. Authors are strongly suggested to provide some visualization to better describe the pruning results, especially considering the semantic guidance mechanism in the design. So does the diagram of the method.
5.  While AdaptiVision shows better FLOPs/accuracy trade-offs, it remains uncertain whether baselines like ToFe-LV and HiRED were re-trained under identical token counts, input resolutions, and data augmentations. FLOPs alone may not fully reflect runtime latency on hardware.

**Questions:**

Please authors elaborate on how AdaptiVision fundamentally differs from existing token-merging or condensation methods such as TokenLearner and DART.

---

### Note · Authors · 2025-12-11

I have read and agree with the venue's withdrawal policy on behalf of myself and my co-authors.